# Surface Grafting of Graphene Flakes with Fluorescent Dyes: A Tailored Functionalization Approach

**DOI:** 10.3390/nano15050329

**Published:** 2025-02-20

**Authors:** Ylea Vlamidis, Carmela Marinelli, Aldo Moscardini, Paolo Faraci, Stefan Heun, Stefano Veronesi

**Affiliations:** 1NEST, Istituto Nanoscienze–CNR and Scuola Normale Superiore, Piazza San Silvestro 12, 56127 Pisa, Italy; stefano.veronesi@nano.cnr.it; 2Department of Physical Science, Earth, and Environment, University of Siena, via Roma 56, 53100 Siena, Italy; carmela.marinelli@unisi.it; 3Scuola Normale Superiore, Laboratorio NEST, Piazza San Silvestro 12, 56127 Pisa, Italy; aldo.moscardini@sns.it (A.M.); paolo.faraci@sns.it (P.F.)

**Keywords:** graphene, fluorescent probes, covalent functionalization, fluorescence microscopy, photoluminescence spectroscopy

## Abstract

The controlled functionalization of graphene is critical for tuning and enhancing its properties, thereby expanding its potential applications. Covalent functionalization offers a deeper tuning of the geometric and electronic structure of graphene compared to non-covalent methods; however, the existing techniques involve side reactions and spatially uncontrolled functionalization, pushing research toward more selective and controlled methods. A promising approach is 1,3-dipolar cycloaddition, successfully utilized with carbon nanotubes. In the present work, this method has been extended to graphene flakes with low defect concentration. A key innovation is the use of a custom-synthesized ylide with a protected amine group (Boc), facilitating subsequent attachment of functional molecules. Indeed, after Boc cleavage, fluorescent dyes (Atto 425, 465, and 633) were covalently linked via NHS ester derivatization. This approach represents a highly selective method of minimizing structural damage. Successful functionalization was demonstrated by Raman spectroscopy, photoluminescence spectroscopy, and confocal microscopy, confirming the effectiveness of the method. This novel approach offers a versatile platform, enabling its use in biological imaging, sensing, and advanced nanodevices. The method paves the way for the development of sensors and devices capable of anchoring a wide range of molecules, including quantum dots and nanoparticles. Therefore, it represents a significant advancement in graphene-based technologies.

## 1. Introduction

Graphene, a single-atom-thick planar sheet of sp^2^-bonded carbon atoms arranged in a honeycomb crystal lattice, has garnered significant attention over the last two decades after its first isolation occurring as recently as 2004 [1]. This groundbreaking discovery has opened up new opportunities for research in physics, chemistry, biotechnology, and materials science. Graphene boasts remarkable properties including exceptional electrical conductivity, mechanical flexibility, optical transparency, thermal conductivity, and low coefficients of thermal expansion [2,3]. These unique characteristics have sparked considerable interest across academia and industry, spanning various fields in the areas of polymer nanocomposites [4], supercapacitor devices [5], drug delivery systems [6], solar cells [7], memory devices [8], field-effect transistor devices [9], biosensors [10], and more. Moreover, the possibility to precisely control the functionalization or chemical derivatization of graphene and graphene oxide (GO) opens new avenues for tailoring and enhancing their properties, thereby expanding their potential applications across various fields [11,12]. Recent advances in the robust and well-controlled functionalization of graphene and graphene oxide have paved the way for a wide range of applications, including aptamer-based biosensors [13], gas sensing [14], and next-generation electrochemical, electronic, and optoelectronic devices [15]. These breakthroughs highlight the immense potential of functionalized graphene in both fundamental research and industrial applications.

In particular, the covalent functionalization of graphene offers several advantages as it enables significantly stronger modifications of the geometric and electronic structure of graphene [16,17]. This method allows to tune the properties of graphene, like enhancing its stability in hydrophilic/hydrophobic media [18,19], combining graphene properties with those of other compound classes possessing specific functions, enabling the modulation of graphene’s molecular-level doping, or systematically tuning the Fermi level by the introduction of electron-donating/electron-withdrawing groups (n-doping or p-doping) [20]. For these reasons, achieving an effective method for producing surface-functionalized graphene sheets on a large scale has become a major objective for many researchers.

The covalent functionalization of graphene, which induces a hybridization change in its carbon atoms from sp^2^ to sp^3^, can be effectively achieved through several reaction mechanisms, with radical and cycloaddition reactions being among the most efficient approaches.

Radical reactions occur through the use of aryl diazonium salts to generate radicals upon electrochemical reduction or through thermal or photoinduced decomposition [21,22]. This functionalization process is triggered by a single-electron transfer from the graphene lattice to the aryl diazonium salt, leading to the formation of an aryl radical via the release of a N_2_ molecule. Cycloaddition reactions provide an alternative and highly effective method of functionalizing graphene, involving the simultaneous formation of two σ bonds with sp^3^ carbons in the graphene lattice. A unique feature of these reactions is that, due to the degeneracy of electronic states at the Dirac point, graphene can act as various synthons (e.g., diene, allyl, etc.), allowing for versatile reactivity in cycloaddition processes [23]. Cycloaddition can be achieved using reactive intermediates such as nitrenes, carbenes, and arynes [24,25,26]. These species covalently modify graphene through CH insertion or cycloaddition reactions. However, this type of reaction, which employs highly reactive intermediates, presents several disadvantages, including the occurrence of undesired side reactions that can disrupt the graphene lattice and the challenge of achieving controlled selectivity, which often leads to spatially uncontrolled functionalization. Additionally, the short lifespan of radical or intermediate species poses a significant challenge as they can decompose rapidly, necessitating precise control over reaction conditions, such as temperature and light, to ensure effective and consistent functionalization.

Currently, one of the most promising routes for the covalent functionalization of graphene is the 1,3-dipolar cycloaddition (1,3-DC) of azomethine ylide, as it is a more selective and controlled method [27]. This approach has been already explored for the chemical modification of carbon nanotubes, fullerenes, and other carbon nanostructures [28,29,30]. The versatility of this method is attributed to the wide range of organic derivatives available through the selection of appropriate precursors. A recent comprehensive investigation has been conducted on the grafting of azomethine ylide onto both graphene nanosheets and reduced graphene oxide via 1,3-dipolar cycloaddition, which included Density Functional Theory (DFT) simulations and investigation of the influence of organic solvents on dispersion properties [31,32].

Here, we demonstrate the possibility to exploit this reaction mechanism to successfully functionalize graphene flakes with specific molecules, such as well-known commercial fluorescence markers for biological and environmental applications [33,34,35]. Small molecule organic fluorescent dyes are primary reagents in scientific research, particularly employed in cells, tissues, microorganisms, and the environment for various applications such as imaging, sensing, and drug delivery [36]. The advantage offered by the incorporation of fluorescent dyes in graphene nanosheets is the direct and fast detection of the functionalized samples by excitation of fluorescence, besides the possibility to localize the most reactive sites (defects) through fluorescence imaging. The literature reports the functionalization of graphene oxide through covalent bonding with fluorescent dyes, exploiting the reactivity of oxygen-containing groups [37]. It is well known that the presence of functional groups, as well as lattice defects such as sp^3^ carbon hybridization, vacancy-like defects, and boundary-like defects, can significantly enhance the reactivity of the carbon atoms in graphene, thus favoring the covalent functionalization with organic molecules [16,38]. Nevertheless, defects in the graphene crystal lattice lead to an undesired degradation of its properties such as electrical conductivity, mechanical strength, and optical characteristics. On the contrary, achieving the functionalization of graphene with low defect density, essential for any application, is still challenging. For this reason, we investigated the covalent functionalization of graphene nanosheets, which, due to their nature, feature low defect density, mainly at the edges.

This work introduces a novel method for the covalent functionalization of graphene flakes with a low defect concentration, marking a significant improvement that enables the attachment of specific molecules such as fluorescence markers commonly used in biological research. A central innovation is the use of a custom-synthesized ylide with a protected amine group tert-butoxycarbonyl (Boc), allowing for further functionalization with target molecules. In contrast to conventional methods that often damage the structure of graphene, this approach minimizes defects while enabling effective functionalization thanks to the high selectivity of the method employed. The first step of functionalization, involving the grafting of the ylide on the graphene flakes via 1,3-DC, was confirmed by Raman spectroscopy. Next, after Boc cleavage, the fluorophores were linked to the ylide. The fluorophores chosen contain a N-hydroxysuccinimide (NHS) ester terminal group, which is frequently employed for the covalent conjugation of amine-containing biomolecules (e.g., proteins, peptides, or to label a protein or peptide) to surfaces through amide linkage [39]. The active ester of the employed fluorophores (Atto 425, Atto 465, and Atto 633 NHS ester) can react with primary amines in slightly alkaline conditions (pH 7.2 to 9) to yield stable amide bonds.

Photoluminescence spectroscopy and confocal microscopy were utilized to detect graphene flakes labeled with fluorescent dyes, enabling the precise characterization of the samples and demonstrating the successful ylide 1,3-dipolar cycloaddition onto graphene, followed by the attachment of fluorescent dyes through NHS ester derivatization.

This approach paves the way for sensor development and device fabrication, providing a flexible platform for anchoring not only fluorescent dyes but also quantum dots, nanoparticles, and other molecules. With broad potential for applications in biological imaging, sensing, and advanced nanodevices, this technique represents a significant advancement in the functionalization of graphene-based materials. In these applications, toxicity is a key point. While some studies suggest a promising biocompatibility of graphene, the toxicity of functionalized graphene remains highly dependent on surface chemistry, functionalization strategy, and potential contaminants [40]. Therefore, further comprehensive studies are essential to ensure its safe and effective use in biomedical applications.

## 2. Materials and Methods

### 2.1. Chemicals

BeDimensional S.p.A. (Italy) supplied the wet-jet milled exfoliated graphene powder (99%). Ethanol (≥96%), phosphoric acid (85%), N,N-dimethylformamide (DFM, anhydrous, 99.8%), N-methylglycine (98%), di-tert-butyl dicarbonate (Boc_2_O, 99%), N,N′-Diisopropylcarbodiimide (DIC, 99%), triethylamine (Et_3_N, >99.5%), hexamethylenediamine (98%), acetonitrile anhydrous (MeCN, 99.8%), dichloromethane (99.8%), methanol (99.8%), anhydrous dimethyl sulfoxide (DMSO, ≥99.9%), 3,4-dihydroxybenzaldehyde (97%), and benzyl-2-bromoacetate (98%) were purchased from Sigma-Aldrich (Darmstadt, Germany). Sodium carbonate (for analysis, ≥99.5%) was purchased from Carlo Erba (Milano, Italy), while Atto 425 NHS ester (>90%), Atto 465 NHS ester (>90%), Atto 633 NHS ester (>90%) were supplied by ATTO-TEC GmbH (Siegen, Germany).

### 2.2. Synthesis of the Linker Molecule

The proposed approach for the functionalization of graphene flakes was implemented through a multi-step process, with a detailed schematic flowchart outlining the key phases presented in Figure 1. This visual representation provides a clear overview of each stage, enhancing the understanding of the methodology.

In this work, the mono-functionalization of diamine is key in holding functional groups at one end while the other end is free to attach to the graphene flakes. Therefore, we synthesized a mono-protected diamine starting from 1,6-hexanediamine and using tert-butyloxycarbonyl (Boc) as an amine protecting group [41].

First, the two amine groups of hexamethylenediamine are differentiated as an acid salt and a free base, which is ready for further functionalization (Figure 1) [42]. Selective protonation of one amine at moderately acidic pH occurs due to inductive and electrostatic effects, ensuring that only one amine forms a salt while the other remains a reactive free base. Protonation of the first amine withdraws electron density along the aliphatic chain, lowering the basicity of the second amine and reducing its likelihood of protonation. Additionally, electrostatic repulsion between two NH_3_^+^ groups further disfavors double protonation under these conditions. Gaseous hydrochloric acid was obtained by a reaction between HCl (0.05 mol) and H_2_SO_4_ (0.05 mol) at 40 °C under a nitrogen atmosphere. HCl was directly bubbled into a flask containing 40 mL of methanol. Hexamethylenediamine (0.05 mol) was dissolved in 20 mL of methanol, and, after 30 min, was slowly added to the reaction flask through a dropping funnel over a 20 min period. The solution was placed in a cooling bath (acetone/ice/NaCl) and stirred overnight. The mono-salified diamine produced from this reaction was dried using a rotary evaporator and subsequently purified by HPLC-MS to eliminate any residual free diamine, ensuring a high-purity final product.

The mono-salified diamine (1.8 mmol) was then dissolved in 40 mL of methanol, and a solution containing Boc_2_O (1 eq.) in 40 mL of methanol was slowly added to let the free amine react, leading to the formation of the mono-Boc protected diamine. Once the addition was complete, the solution was stirred for about 1 h in a cooling bath under a nitrogen atmosphere. This method has demonstrated high efficiency in contrast to the conventional approach which requires careful control over reagents concentration and a gradual addition of Boc_2_O solution over a long period of time to prevent di-Boc formation [43,44]. Once the reaction was completed, the solvent was evaporated, and diethyl ether was added to remove the unreacted diamine residue. Then, 1M NaOH solution was added to obtain the amine salt free. The reaction product was then extracted by organic solvent, washed in dichloromethane, and dried using rotary evaporator equipment.

The mono-Boc diamine (6 mmol) was then dissolved in 40 mL of anhydrous acetonitrile in a three-neck round-bottomed flask equipped with a magnetic stirring bar. Then, a solution of benzyl-2-bromoacetate (alpha-bromo derivative, 2 mmol) and trimethylamine (4 mmol) in anhydrous acetonitrile (20 mL) was slowly added to the reaction flask through a dropping funnel. The reaction was carried out under nitrogen atmosphere, and after 30 min, the reaction was complete. Finally, the product was analyzed and purified using a tandem UHPLC-MS system (refer to Appendix A) before it was freeze-dried and stored at −20 °C.

### 2.3. Covalent Functionalization of Graphene Flakes by 1,3-Dipolar Cycloaddition

The first step involves the covalent functionalization between the organic linker and the graphene flakes, as described in Figure 2. For this reaction, 3,4-dihydroxybenzaldehyde (6 mg, 0.18 mmol), sodium carbonate (5 mg), and the previously synthesized linker molecule (6 mg, 0.28 mmol) were added to the few-layers graphene suspension (2 mg) in DMF. Specifically, sodium carbonate (which reduces the acidity of the reaction environment) and 3,4-dihydroxybenzaldehyde are necessary for the in-situ formation of the 1,3-dipolar compound (azomethine ylide), which ultimately binds to graphene through the formation of a pyrrolidine ring (refer to Figure 2). In the reaction mechanism, the formation of the imine (reaction intermediate) increases the acidity of the alpha proton, leading to the formation of a carbanion (negatively charged carbon atom). The shift in the double bond, in turn, results in the formation of the 1,3-dipoles that represent the reactive species.

The reaction was carried out for 120 h at 150 °C under magnetic stirring [31,32]. Fresh excess reagents were added daily to drive the reaction and increase the yields of graphene functionalization, as the ylide is susceptible to deactivation through side reactions with 3,4-dihydroxybenzaldehyde or even through self-reactivity with another ylide molecule. To avoid secondary reactions resulting from solvent oxidation at elevated temperatures, the reaction was conducted under a nitrogen atmosphere. The resulting suspension was then washed several times with clean solvent. The mixture was centrifuged at 13,000 rpm, the solvent was removed, clean solvent was added, and then the flakes were re-suspended by sonication. This procedure was repeated several times using DMF, water, acetone, and eventually chloroform.

The second step involves the deprotection of the amine group. Conventional methods for N-Boc deprotection primarily rely on cleavage in acidic conditions [45]. The Boc protecting group was cleaved, suspending the graphene flakes functionalized with the linker in dichloromethane (600 μL), and then phosphoric acid (15 eq.) was added. The mixture was stirred for about 6 h at room temperature and checked periodically by FT-IR spectroscopy to confirm the completion of the reaction (refer to Appendix A).

Eventually, the flakes were rinsed in dichloromethane and methanol by sequentially centrifugation and resuspension.

Once the amine group was deprotected, the fluorophore was linked to the amine. This third reaction step was carried out suspending the graphene flakes-linker in 500 μL of DMSO containing the desired fluorophore (10 μg). In the reaction ambient, triethylamine was added to reach a mild alkaline pH, and DIC (100 μL) was used to promote amidation reaction by the activation of the carboxyl group [46]. The mixture was stirred overnight at room temperature, and the reaction flask was carefully protected from light. The graphene flakes were rinsed in DMSO, acetonitrile, and acetone to remove the excess of reagents and catalyst. The fluorophore-labeled graphene flakes were dried and stored at −20 °C in a dark vial bottle.

### 2.4. Spectroscopic, Optical, and Morphological Characterization Techniques

For the spectroscopic, optical, and morphological characterization, 5 μL of the graphene flakes suspended in chloroform were applied onto clean silicon substrates with native oxide (Si/SiO_2_) by the drop-casting method and dried under nitrogen flux.

Raman spectroscopy was performed using a Renishaw InVia system featuring a confocal microscope, an excitation laser with a wavelength of 532 nm (2.62 eV), and an 1800 L/mm grating (spectral resolution of 2 cm^−1^). All spectra were acquired with a 100× objective (NA = 0.85, spot size ~1 µm) using the following parameters: excitation laser power ~90 mJ μm^−2^, two acquisitions per spectrum. A comprehensive statistical analysis was performed on 75 Raman spectra collected from the surfaces of different graphene flakes, both for pristine and functionalized graphene, providing robust insight into the uniformity and consistency of the functionalization.

Photoluminescence spectra were acquired with an excitation wavelength of 473 nm using the Raman spectrometer, with 2400 L/mm grating, 50× objective, and ~6 mJ μm^−2^ laser power.

Fluorescence imaging and lifetime measurements were performed with a confocal microscope (Leica Microsystems, Wetzlar, Germany), equipped with a 63×, oil immersion objective (Leica Microsystems). Pulsed diode lasers operating at a frequency of 40 MHz were employed for excitation, and each sample was excited as close as possible to the wavelength of maximum absorption of the fluorophore molecules under investigation. Fluorescence lifetime imaging microscopy (FLIM) was employed to provide insights into the local microenvironment and molecular interactions, as fluorescence decay patterns are significantly influenced by factors such as local pH changes, energy transfer, and molecular binding [47].

Fluorescence microscopy images were analyzed using ImageJ software. For each image, the color of the pixels corresponds to the perceived color of the fluorescence emission (spectral emission wavelength).

The atomic force microscopy (AFM) data were acquired with a Bruker Dimension Icon system, operating in tapping mode, and Gwyddion software was used for processing the images.

Scanning electron microscopy (SEM) images were acquired with a Jeol JSM-7500F instrument using the following parameters: working distance of 6 mm, acceleration voltage of 5 kV, and emission current of 20 μA.

## 3. Results and Discussion

### 3.1. Surface Characterization

Graphene flakes produced by wet-jet milling [48] were dispersed in DMF by homogenization in order to obtain a stable dispersion (~0.2 mg mL^−1^) [31]. Such graphene dispersion exhibits stacks of several overlapping layers with a smooth planar structure, showing some scrolling on the edges of the graphene (refer to AFM and SEM characterizations in Figure 2). According to the morphological characterization, the nanosheets display different lateral sizes, ranging from a few hundred nanometers to ~3 μm. In the AFM profile obtained from a flat area, heights corresponding to a few layers of graphene can be identified, with thickness ranges from about 0.6 to 1 nm (Figure 2b).

### 3.2. Raman Characterization of Graphene Functionalized with the Linker by 1,3-Dipolar Cycloaddition

The first step of functionalization, i.e., the grafting of the linker molecule on graphene flakes by cycloaddition, was confirmed by Raman analysis before and after the cycloaddition. Figure 3 shows representative Raman spectra of the pristine graphene flakes compared to those of the graphene-linker sample, along with a typical optical image. The typical features in the Raman spectrum of graphene are the G band and the 2D band, centered at 1581.6 cm^−1^ and 2720.8 cm^−1^, respectively. The 2D band shows a full width at a half maximum (FWHM) of about 64 cm^−1^, which allows us to identify the graphene flakes as few- layer graphene [49]. The D peak, centered at 1350.5 cm^−1^, and the D′ peak, here centered at 1616 cm^−1^, indicate the presence of defects in the lattice, which breaks the symmetry of the carbon honeycomb lattice. For low defect concentrations, the D peak intensity is proportional to the amount of defects [50]. For pristine graphene, the ratio I(D)/I(D′) is ~4.8, suggesting the presence of boundary-like and vacancy-like defects in the sample [51].

The Raman spectrum of functionalized graphene (Figure 3a) shows clearly a decreased intensity of the D peak compared to the pristine sample. The defect density can be assessed by the peak intensity ratio between the D band and the G band. The inset of Figure 3a presents the statistical analysis of 75 Raman spectra acquired on the surfaces of different few-layer graphene flakes of both pristine and functionalized samples, providing a detailed comparison of the defect density. Here, the ratio I(D)/I(G) varies from an initial average value of ~0.11 for the pristine sample to a value of ~0.022 for functionalized graphene flakes. This decrease in I(D)/I(G) can be explained considering that the linker molecule grafts onto graphene’s most favorable bonding sites, which are the lattice defects, possibly leading to a local structural relaxation and a decrease in the Raman intensity of the defects. Considering that the lateral size of the graphene flakes is comparable with the Raman laser spot size (~1 µm), edge defects contribute significantly to the intensity of the D peak of the pristine graphene flakes, which, after functionalization, are saturated by the presence of the molecules, leading to a decrease in the intensity of the D peak [31,52]. On the contrary, the intensity and the position of the 2D band are not greatly affected compared to pristine graphene, confirming that the process does not significantly modify the structural order of the graphene lattice, and no disorder due to amorphous phases was introduced [49].

The degree of functionalization was investigated in a previous study involving graphene flakes functionalized using the same reaction mechanism and parameters—with a closely similar azomethine ylide—which demonstrated a functionalization efficiency of approximately one ylide per ~170 carbon atoms, as estimated by X-ray photoelectron spectroscopy (XPS) analysis [31]. Therefore, it is reasonable to assume a comparable functionalization rate in this case as well.

### 3.3. Optical Characterization of Graphene Flakes Functionalized with the Fluorophores

After the deprotection of the amine group and the reaction with the fluorophore Atto 465 NHS, photoluminescence (PL) spectroscopy was employed to confirm the successful functionalization of the flakes.

Frequently, for many compounds such as carbon nanomaterials, auto-fluorescence arises upon excitation in the green spectrum range [36]. For this reason, the optical characterization of a reference sample consisting of pristine graphene flakes was acquired. As demonstrated in Figure 4a, the PL spectrum of the pristine graphene flakes shows a negligible intrinsic fluorescence compared to the emission of the Atto 465-labeled graphene flakes, which is characterized by a broad band centered at about 520 nm, typical of this fluorophore (see Appendix A). In addition to the fluorescence, using a laser power of 6 mJ µm^−2^, the bands associated with electronic transitions involving graphene are also evident at ~505 nm, 511 nm, and 543 nm. The corresponding Raman shifts are calculated considering the excitation wavelength used (473 nm) and yield values of approximately 1340 cm^−1^, 1572 cm^−1^, and 2725 cm^−1^, thus ascribable to D, G, and 2D graphene bands, respectively. Furthermore, it is noteworthy that for laser energies exceeding ~12 mJ μm^−2^, photo-bleaching was observed due to chemical damage and covalent modification [53].

To exclude both a possible signal arising from the graphene-linker system and the possibility that the fluorescent dye bonds directly to the graphene flakes, two more samples were prepared and optically characterized. One sample consisted of graphene flakes functionalized only with the linker molecule (performing only the first step of functionalization), and the second was obtained by directly performing the last reaction step between the pristine graphene flakes (without linker) and the fluorophore Atto 465 NHS under the conditions described in the Experimental Section. The PL spectra of the samples are presented in Figure 4b. Graphene flakes functionalized solely with the linker exhibit a PL spectrum similar to that of the pristine sample, showing no significant fluorescence signal (see Figure 4b). Also, the sample prepared without the initial cycloaddition step shows no detectable PL bands. This strongly suggests that no direct interaction occurs between the graphene flakes and the fluorophore, further confirming the absence of physisorption of the dye molecule onto the graphene surface.

After confirming that the PL signal was only observed when the fluorophore was chemically bound to graphene via the linker molecule, the Atto 465-labeled graphene flakes sample was further characterized using a confocal fluorescent microscope. At low magnification (5× objective), when the flakes on Si/SiO_2_ are irradiated with blue light, they appear as uniformly distributed bright spots emitting light, as shown in Appendix A. Under excitation at 470 nm, which corresponds to the absorption maximum for the fluorophore, the emission spectra show a broad curve (FWHM ~120 nm) with a maximum at ~510 nm, as expected for this fluorophore (Figure 4c) [34]. This fluorescence profile reveals that the molecules do not undergo evident modifications upon graphene linkage. Compared to the functionalized sample, the pristine graphene flakes show a negligible emission due to a mild autofluorescence phenomenon at the set wavelength (Figure 4c). In addition, analysis of the fluorescent image (inset of Figure 4c) and the spectra in Appendix A clearly indicates that the luminescence intensity—corresponding to the degree of surface functionalization—is relatively uniform across the graphene flakes. However, variations in luminescence intensity are observed across different ROIs, likely due to the random distribution of defects. While defects are expected to be more concentrated at the edges of graphene flakes [54], where covalent bonds readily form due to higher reactivity [55], the applied functionalization method effectively achieves a uniform derivatization of both the edges and the basal plane. Such homogeneous modification ensures consistent chemical interaction throughout the flakes, a critical factor for reproducibility and the optimization of material properties in various applications. Furthermore, the observed homogeneity in the fluorescence intensity distribution closely matches the nitrogen atom distribution determined in our prior work using energy-dispersive X-ray spectroscopy (EDX) following the grafting of a similar ylide onto graphene [31].

Along with the emission spectra, FLIM images of the graphene flakes were acquired to investigate the decay curves and the fluorescence lifetime. Due to the significant difference in fluorescence intensities observed between the functionalized and control samples, the curves were normalized for easier comparison. To further characterize the photophysical properties of the samples, the time-resolved room temperature decay curves of both the pure fluorophore and the Atto 465-labeled graphene samples at the same excitation wavelength of 470 nm were investigated, and the corresponding measurement results are presented in Figure 4d. The distinct decay profiles observed for these two emissions reveal different excited state dynamics, highlighting the different photophysical behavior of the fluorophore when interacting with graphene. The pure fluorophore Atto 465 dissolved in chloroform displays, as expected, a mono-exponential decay with a lifetime of ~3.5 ns [56,57]. On the contrary, Atto 465-labeled graphene flakes reveal a decay curve which can be fitted by a double exponential function, with estimated average lifetimes of τ_1_ = 0.8 ns, and τ_2_ = 2.9 ns (refer to Appendix A for further details).

Solutions of free fluorophores typically undergo a first-order decay process, governed by a single dominant relaxation pathway, leading to a single-exponential fluorescence decay. However, when these fluorophores are anchored to surfaces such as graphene or other substrates, their fluorescence decay behavior can change significantly. Multi-exponential decay patterns are commonly observed for fluorophores attached to molecules like nucleosides or cellular membranes [58,59], indicating that the fluorophores undergo multiple decay pathways due to various interactions with their environment.

In the case of Atto 465 anchored to graphene, the observed double-exponential decay curves can be attributed to multiple energy transfer processes occurring at the dye-substrate interface, including energy transfer to the graphene’s conjugated π-system and charge interactions with the substrate [60]. This introduces a dual decay process: one representing rapid energy transfer to graphene and another corresponding to the intrinsic fluorescence decay of the fluorophore itself. Thus, the attachment to graphene alters the fluorescence dynamics by adding new decay channels, which reflects the complexity of interactions at the dye-substrate interface.

Moreover, the substrate can impose physical constraints on the fluorophores, affecting their rotational or conformational freedom, thereby influencing their radiative and non-radiative relaxation rates. The fluorophores’ immobilization in a more rigid environment typically reduces non-radiative decay pathways, leading to an increase in fluorescence lifetime [61]. These micro-environmental factors contribute to the observed multi-exponential decay patterns, which reflect the combined effects of energy transfer, charge interactions, and molecular constraints on the fluorophores. In this regard, we observed that the lifetime values also could vary slightly considering different ROIs on the sample. For this reason, the lifetimes reported for the functionalized graphene samples are average values calculated considering several areas in the FLIM images.

Having confirmed that graphene flakes can indeed be labeled with dyes using the above-described procedure, we continued to label and characterize graphene flakes with other fluorescent dyes. In Figure 5, panel a, the optical characterization and fluorescence imaging of graphene flakes-Atto 425 NHS are shown. The emission spectrum of the sample was acquired using a confocal microscope under a 440 nm excitation wavelength (absorption maximum) [29]. The acquired spectrum, reported in Figure 5a, shows an emission curve with a maximum around 485 nm, comparable with the values reported in the literature for this fluorophore. As already observed under the previous excitation wavelength, negligible fluorescence was detected from the pristine sample.

To definitely exclude any possible interference due to autofluorescence phenomena from the samples, in a further experiment, the graphene flakes were functionalized with a fluorophore emitting in the red spectrum range, namely, Atto 633 NHS. As displayed in Figure 5b, upon excitation at 640 nm, the sample shows a fluorescence spectrum peaking around 652 nm, thus matching the literature data [29], while almost no fluorescence signal was detected from the control sample (pristine graphene flakes).

Once again, fluorescence imaging of these two samples (insets of Figure 5a,b) reveals a well-distributed functionalization across the entire surface, highlighting the uniformity of the surface functionalization. This observation further validates the effectiveness of the proposed process, confirming the consistent spatial distribution of molecules and highlighting the reliability of the applied modification strategy.

Consistent with the previous results, graphene flakes labeled with both Atto 425 and Atto 633 exhibited decay curves similar to those observed using Atto 465 (refer to Appendix A). The lifetime values obtained for each functionalized sample, as well as for each free fluorophore in solution, are reported in Appendix A.

Besides the possibility to use the linker molecule as a platform to label graphene with a variety of molecules, another compelling aspect of the graphene samples investigated is that after storage in ambient conditions, protected from direct light exposure, they were found to be stable for over a year, i.e., over time they retain the ability to emit upon absorption at a proper wavelength.

## 4. Conclusions

In this work, we presented a simple mechanism to successfully functionalize low-defect graphene flakes with fluorescent dyes through the linkage of an organic molecule. The functionalization reaction was carried out in three steps involving (I) the 1,3-dipolar cycloaddition of a custom-synthesized azomethine ylide to graphene, (II) deprotection of a terminal amine group, and (III) chemical binding of a fluorophore to the linker molecule using NHS ester derivatization. The fluorescence spectra acquired from the samples labeled with three different fluorophores (Atto 425, Atto 465, and Atto 633 NHS ester) showed the typical emission curves of the dyes, demonstrating the effectiveness of the procedure. This proof-of-concept experiment sets the stage for exploiting linker molecules attached to graphene in the development of platforms for various applications, thanks to the possibility to subsequently attach molecules or functional groups to the linkers. The potential to functionalize low-defect graphene systems through this approach enables the design of customized devices, where the unique properties of graphene can be synergistically combined with those of other molecules, nanoparticles, or quantum dots by a straightforward and easily controllable mechanism.

## Data Availability

The original contributions presented in this study are included in the article/Appendix A. Further inquiries can be directed to the corresponding author.

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
