# Peer review of "Surface Grafting of Graphene Flakes with Fluorescent Dyes: A Tailored Functionalization Approach"

_nanomaterials, 2025, doi:10.3390/nano15050329_

Round 1
Reviewer 1 Report
Comments and Suggestions for Authors
The manuscript of Vladimis Y. et al. displays the results on the covalent modification of graphene flakes with the fluorescent dyes (Atto 425, 465, and 633) through 1,3-dipolar cycloaddition. The novel technique derived from employing custom-synthesized ylide with a protected amine group (Boc), facilitating subsequent attachment of functional molecules, is demonstrated. Overall, a decent work of practical interest, which is done on a good scholarly level. The manuscript is suitable for being published in the Nanomaterials journal but after Major revision to address the queries below
1. The introduction section should be refined, adding recent publications on the covalent modification of graphene and its effect on the physical properties, particularly: 10.1021/acsami.3c02833, 10.1038/s42254-022-00422-w, 10.3390/nano11071717
2. The synthesized graphene derivative is thoroughly examined by photoluminescence spectroscopy, fluorescence imaging, and lifetime measurements, which verify the successful covalent modification of the graphene flakes, employing the proposed method. However, neither of these methods gives a hint about the functionalization rate. The authors should additionally probe the synthesized materials via X-ray photoelectron spectroscopy (XPS) or, at least, Elemental analysis (demonstrating the presence of nitrogen in the modified samples) and provide the functionalization degree that is achieved.
3. From the text it remains unclear what is the factor behind only one amine group of hexamethylenediamine becoming a salt and the other remaining a free base. Since it is one of the key points of the performed modification, authors should clarify this.
4. The authors are encouraged to emphasize the fact that the performed modification results in a uniform derivatization of both edges and the basal plane of the graphene flakes as seen from the fluorescence imaging in Figures 4 and 5. In the current version of the text, it is repeatedly mentioned that the edges would be predominantly functionalized, which is reasonable. However, the successful derivatization of the basal plane is more important and since it is achieved in this work it should be emphasized.
5. Following the previous query, it is interesting how the modification affects the electronic structure and, particularly, the graphene's п-conjugation degree. The detailed examination of these parameters is out of the scope of this article. However, the authors are encouraged to probe them through UV-Vis studies, analyzing the changes in the absorbance spectra, namely the integral intensity of the optical absorption in the Visible region. Despite the effect of the dye absorption bands, the presence/absence of the overall diminishment in the absorption and its more pronounced dependence on the wavelength will give a hint on the level of the disruption of the п-conjugation due to the covalent modification (see, 10.3390/nano11040915).
Reviewer 2 Report
Comments and Suggestions for Authors
Reviewer report on manuscript nanomaterials-3466312
Ylea Vlamidis et al. “Surface grafting of graphene flakes with fluorescent dyes: a tailored functionalization approach”
In present study, method of 1,3-dipolar cycloaddition has been extended to graphene flakes with low defect concentration. A novelty is the use of a custom-synthesized ylide with a protected amine group (Boc), facilitating subsequent attachment of functional molecules. Indeed, after Boc cleavage, fluorescent dyes (Atto 425, 465, and 633) were covalently linked via NHS ester derivatization. This approach represents a highly selective method minimizing structural damage. The successful functionalization was demonstrated by Raman spectroscopy, photoluminescence spectroscopy and confocal microscopy, confirming the effectiveness of the method.
The manuscript can be accepted after minor revision.
Overall, the quality of the work is good, however I point out several questions to help the authors improve the manuscript before publication.
Questions/comments:
1. The introduction doesn’t include all relevant up-to-date references. Some important up to date (2024) references in this field should be added, e.g. [Carbon 2022, 196, 264], and references there.
2. The Raman spectra identification is not very good justified. There are not up-to-date references (2024) for choice of the components. The references to up-to date 2024 investigations in this field should be added. For such identification, I recommend using up-to-date publication [Carbon, 2022, 194, 52-61], and references there.
- Typos should be corrected., e.g.
Page 8, Line 297: Should be “of defects [45].” instead of “of defects.45”

Reviewer 3 Report
Comments and Suggestions for Authors
This manuscript reported a novel method for the functionalization of graphene with fluorescent dyes by using 1,3-dipolar cycloaddition with the use of a custom-synthesized ylide with a protected amine group. This work is novel and could help for the development of sensors and devices capable of anchoring a wide range of molecules.
Some comments:
1. How about the yield of the dye-modified graphen?
2. Why Atto 425 and Atto 633 anchored on grapheme exhibited double exponential fluorescence decay curves, whereas the free fluorophores showed the typical first order reactions in solution?
3.What’s the performance such as life time of these anchored Atto 425 and Atto 633 compared to the immobilization of them on other substrates?
4. Since the method is positioned for biological imaging and sensing, a brief discussion of potential challenges and considerations regarding the biological compatibility and toxicity of the functionalized graphene would be helpful.
Round 2
Reviewer 1 Report
Comments and Suggestions for Authors
The authors have adequately answered all the queries. The manuscript is now suitable for being published in its current form